# Knowledge and awareness-based survey of COVID-19 within the eye care profession in Nepal: Misinformation is hiding the truth

**Sandip Das Sanyam**[1]*, **Sanjay Kumar Sah**[2], **Pankaj Chaudhary**[1], **Matthew J. Burton**[3], **Jeremy J. Hoffman**[1,3]

**1** Sagarmatha Choudhary Eye Hospital, Lahan, Nepal, **2** Biratnagar Eye Hospital, Biratnagar, Nepal, **3** International Centre for Eye Health, London School of Hygiene and Tropical Medicine, London, United Kingdom

* dassandiip@gmail.com

## Abstract

### Background

Nepal was under a severe lockdown for several months in 2020 due to the COVID-19 pandemic. There were concerns regarding misinformation circulating on social media. This study aimed to analyse the knowledge and awareness of COVID-19 amongst eye care professionals in Nepal during the first wave of the pandemic.

### Methodology

We invited 600 participants from 12 ophthalmic centres across Nepal to complete a qualitative, anonymous online survey. Altogether, 25 questions (both open and closed-ended) were used. An overall performance score was calculated from the average of the 12 "Knowledge" questions for all the participants.

### Results

Of the 600 eye care professionals invited, 310 (51%) participated in the survey. The symptoms of COVID-19 were known to 94%, whilst only 49% of the participants were aware how the disease was transmitted, with 54% aware that anyone can be infected with SARS-CoV-2. Almost 98% of participants recognized the World Health Organization's (WHO) awareness message, but surprisingly, 41% of participants felt that consumption of hot drinks helps to destroy the virus, in contradiction to WHO information. Importantly, 95% of the participants were aware of personal protective equipment (PPE) and what the acronym stands for. Social distancing was felt to be key to limiting the disease spread; whilst 41% disagreed that PPE should be mandatory for eye care practitioners. The mean overall "Knowledge" performance score was 69.65% (SD ± 22.81).

### Conclusion

There is still considerable scope to improve the knowledge of COVID-19 amongst ophthalmic professionals in Nepal. Opinion is also split on measures to prevent transmission, with

**Data Availability Statement:** The data underlying the results presented in the study are available at https://doi.org/10.5061/dryad.bcc2fqzbx.

**Funding:** MJB and JH are supported by the Wellcome Trust (207472/Z/17/Z).

**Competing interests:** The authors have declared that no competing interests exist.

misinformation potentially fuelling confusion. It is recommended to follow WHO and national guidelines, whilst seeking published scientific evidence behind any unofficial statements, to accurately inform one's clinical practice.

## Introduction

The first case of COVID-19 in Nepal was reported on 13[th] January 2020 [1]; at the time of publication, the total count in Nepal is over 275,000. In late 2019, an increase in cases of pneumonia of unknown aetiology in Wuhan, Hubei Province in China, prompted intense research [2]. The Center for Disease Control and prevention, China, extracted and identified the causative virus as a unique coronavirus. Following this, the World Health Organization (WHO) named this new pathogen the 2019-novel-Coronavirus [3], and suggested on 25[th] Dec 2019 that the virus is less contagious to healthcare professionals compared to other recent coronaviruses responsible for severe acute respiratory syndrome (SARS) and the Middle East respiratory syndrome (MERS) [4]. The virus was subsequently renamed as SARS-CoV-2 by the International Committee of Taxonomy of Viruses (ICTV) [5], with the disease it causes named coronavirus disease 2019 (COVID-19). On 31 January 2020, the WHO characterized the outbreak as a public health emergency of international concern [6].

Although the exact origin of the virus still remains elusive, the natural host of the virus is believed to be a species of bat [7]. Initially, it was thought to be transmitted only as a zoonotic infection from animal to human, but subsequent research demonstrated that SARS-CoV-2 is highly contagious and can rapidly transmit between people through respiratory droplets ($> 5$–10 μm in diameter) and fomites [2, 8–13]. The virus can remain viable on different surfaces, which may become a source of infection to individuals if hands are not washed properly [14, 15].

COVID-19 varies in severity from being asymptomatic to causing respiratory failure that necessitates mechanical ventilation and support in an intensive care unit (ICU). Severe disease can lead to multiorgan failure, coma, and death [16]. COVID-19 has been previously reported to be associated with conjunctivitis in humans [17].

Developing countries such as Nepal, with limited healthcare infrastructure and capacity, may struggle to cope if the pandemic follows a similar trajectory to other countries. The Government of Nepal implemented a nationwide "lockdown" since the declaration of the pandemic on 23[rd] March 2020, with severe limitations on movement of people. During this period, there were rumours and misinformation on COVID-19 emerging, which might mislead ophthalmic personnel.

Dr Li Wenliang from Wuhan, who died from COVID-19, was an ophthalmologist who warned about the contagious nature of this virus and had asked eye care practitioners worldwide to take safety precautions, given the close proximity eye care practitioners are in with their patients [18]. The aerosols generated from diseased patients attending for eye examinations have a higher tendency to infect eye care personnel if the appropriate protection is lacking. A pilot study of ours using the same study tool revealed poor knowledge and awareness within the eye care practitioners at our institution. The aim of this study was to review the knowledge and awareness of COVID-19 amongst eye care personnel in Nepal. If gaps in knowledge were found, then this could be addressed through targeted education and training, which will help reduce the risk to healthcare workers and promote good infection control practices.

## Methodology

We performed a qualitative approach to quantify knowledge and awareness of COVID-19 among eye care personnel working in Nepal. This study conformed to the tenets of the Declaration of Helsinki, with a mandatory informed written consent taken digitally prior to participation. There were no patients or minors involved in this study; ethical permission was obtained from Ethical Review Board (ERB) of Sagarmatha Choudhary Eye Hospital, Ref 49-2077/078, dated 1st of March 2020.

The research was carried out as an online questionnaire completed between 28th March to 18th April 2020. Nepal has approximately 1600 eye care professionals of different cadres (ophthalmologists, optometrists, ophthalmic assistants, and other ophthalmic personnel). The survey was sent to 600 of these clinicians working in Nepal. Participation was anonymous and voluntary. Representatives from a random selection of 12 of the 18 secondary or tertiary eye hospitals and clinics across the spectrum of health facilities within Nepal were contacted and asked to select 50 individuals randomly at their institution and contacts who were involved in eye care practice during COVID-19, who were then invited to take the survey after their agreement. This random sample of 600 ophthalmic clinicians (approximately 40% of eye care professionals within Nepal) was calculated to be representative of the staff involved in eye care, whilst being practical enough to implement, making use of established networks between eye care providers.

We used qualitative methods to generate the questions on knowledge and awareness. This has been shown to enrich the quality of questionnaire items and to improve the content validity of the questionnaire [19]. Questions developed in this way derives item generation from the population of interest, rather than that of the researchers. Focus group discussion took place between eye care professionals, statisticians, and microbiologists, facilitated by the investigators. The survey was divided into two sections: section 1 –demographic information; section 2–25 open- and closed-ended questions relating to knowledge and awareness of COVID-19. We implemented a non-probability convenience sampling method to reach the study population through social media applications (WhatsApp, Facebook, Telegram, Email and Viber). Each participant had only one chance to take the survey and two reminders were sent for individuals to complete the survey. In order to ensure validity, online forms could only be completed by those directly invited and values were entered directly by the individual. The responses were anonymous, with no participant-identifiable questions. Data were exported into MS Excel. An overall performance score was calculated from the mean of the 12 Knowledge questions for all the participants. These were categorised as "excellent" (if the average of correct responses was 80% or above), "satisfactory" (if the average of correct responses was 65% or above), or "poor" (if the average of correct responses was below 55%), in a similar way to academic grading. Fisher's exact testing was performed to look for significant associations between categorical variables.

## Results

The survey was completed by 310 (51%) individuals (55.5% male). The majority of participants (83.9%) were in age group 20–30. The demographic details are presented in Table 1. These eye health personnel were from a diverse range of health facilities, from small single-practitioner clinics to stand-alone tertiary referral eye hospitals.

The responses to questions are given in Table 2.

Of the participants, 76% said that knowledge of COVID-19 is essential to eye care practitioners. Despite this, there was a significant variation in the percentage of questions answered correctly. The symptoms of COVID-19 were known to 94%. However, only 54% were aware that anyone can be infected with SARS-CoV-2 and only 49% of the participants were aware of how COVID-19 is spread. Only 5% of the participants did not know the abbreviation of PPE

**Table 1. Demographic details of participants.**

| Demographics | N | (%) |
|---|---|---|
| Survey Completion | 310 | (51.0) |
| Age group 20–30 | 260 | (83.9) |
| Age group 31–40 | 44 | (14.2) |
| Age group 41–50 | 3 | (1.94) |
| Optometrists | 196 | (63.2) |
| Ophthalmic assistants | 84 | (27.1) |
| Ophthalmologists | 17 | (5.50) |
| Other ophthalmic personnel | 13 | (4.20) |

(personal protective equipment) and 17% were unaware of the abbreviation RDT (rapid diagnostic tests). Regarding how COVID-19 is diagnosed, 56% responded correctly that Reverse Transcriptase-PCR for SARS-CoV-2 is used. Almost 98% of participants recognised the WHO's awareness message, although 41% of participants believed that consumption of hot drinks kills the virus (in contradiction to information from the WHO). Overall, 80% of the participants felt that social distancing would be the key to restrict the disease spread.

Almost half of the participants were not sure regarding the life span of the virus on different surfaces, whilst 41% disagreed on a point that PPE is should be mandatory for eye care practitioners. About 20% of the participants were not aware on the correct maintenance of a mask, which is an essential component of PPE. Most participants had PPE to work under COVID-19 crisis. The majority of participants were aware that washing hands with soap and water or alcohol-based sanitiser for at least 20 seconds kills viruses. Only 41% of the participants were able to answer that the multipurpose disinfectant used in clinics was 70% ethanol. Only 45% of the participants said that they follow WHO COVID-19 guidelines.

Interestingly, 94% of the participants knew that the most reported ocular manifestation of SARS-CoV-2 is conjunctivitis. Despite being ophthalmic clinicians, 30% were unable to identify what constituted an ocular emergency. However, 82% of the participants believed that diagnostic testing like tonometry, ocular coherence tomography, etc. must be postponed.

Altogether 81.9% (n = 254) of participants agreed with the statement 'Avoid investigations in ophthalmology such as tonometry, OCT, refraction, etc. until this pandemic resolves.' Descriptive reasons for allowing routine ophthalmic testing to continue were grouped into three categories: 1) if necessary, in case of emergency 19/56 (33.93%); 2) perform only with COVID-19 infection control measures to aid in diagnosis 26/56 (46.43%); and 3) should perform as usual for routine attendance regardless of the COVID-19 situation 11/56 (19.64%).

The free text question 'Being an eye care practitioner, what do you think is wise to do during this period of lockdown?' The descriptive answers were categorised into different sub-categories, illustrated in Fig 1.

The calculated mean knowledge performance score was 69.65% (SD ± 22.81), falling into the "satisfactory" category.

We found no evidence to suggest an association between demographic variables (specifically age, gender and occupation) and answering the knowledge questions correctly (data not shown).

## Discussion

In Nepal, COVID-19 has been in the headlines since the beginning of 2020, and at the forefront of people's minds since the lockdown began in March 2020. Currently, there are sporadic

**Table 2. Knowledge and awareness questionnaire.**

| KNOWLEDGE | Questionnaire | Options (n) % | | | | | | |
|---|---|---|---|---|---|---|---|---|
| | | **A** | **B** | **C** | **D** | **E** | **F** | **G** |
| | What does PPE stand for? | **(293) 94.52** | (12) 3.7 | (05) 1.61 | | | | |
| | What does RDT stand for? | (10) 3.22 | **(259) 83.55** | (36) 11.62 | (05) 1.61 | | | |
| | What is the confirmatory test for SARS–COV2? | (02) 0.65 | (72) 23.23 | **(175) 56.45** | (24) 7.74 | (37) 11.93 | | |
| | Consuming hot fluids, garlic/ginger mix., Vitamin C and soup reliably kills the virus. Do you agree? | (132) 42.58 | **(171) 55.16** | (07) 2.26 | | | | |
| | How long does it take the virus to die from a contaminated surface? | (55) 17.74 | (25) 8.06 | (14) 4.52 | **(83) 26.77** | (133) 42.1 | | |
| | What is the appropriate time for handwashing? | (50) 16.13 | (18) 5.82 | **(211) 68.06** | (03) 0.96 | (28) 9.03 | | |
| | Which are the most effective disinfectants for use in ophthalmic clinics? | (56) 18.06 | **(127) 40.97** | (34) 10.97 | (13) 4.19 | (11) 3.55 | (14) 4.52 | (55) 17.74 |
| | Which one of these eye diseases is related to COVID-19? | **(292) 94.2** | (03) 0.96 | (01) 0.32 | (01) 0.32 | (13) 4.19 | | |
| | Which of the following constitutes and ophthalmic emergency? | (15) 4.84 | (29) 9.35 | (30) 9.68 | (17) 5.48 | **(290) 70.65** | | |
| | What are the symptoms of COVID-19? | (07) 2.26 | (03) 0.97 | (08) 2.58 | (01) 0.32 | **(291) 93.87** | | |
| | Avoid investigations in Ophthalmology like tonometry, OCT, refraction, etc. until this pandemic resolves. Do you agree? | **(254) 81.94** | (56) 18.06 | | | | | |
| AWARENESS | What do you think would be best to reduce the mortality rate caused by COVID-19? | (250) 80.65 | (08) 2.58 | **(12) 3.87** | (34) 10.97 | (06) 1.93 | | |
| | How important is knowledge of COVID-19 for eye care practitioners? | **(238) 76.77** | (58) 18.71 | (14) 4.52 | | | | |
| | How are you helping the community combat this pandemic? | (85) 27.42 | (24) 7.74 | **(141) 45.48** | (41) 13.23 | (19) 6.13 | | |
| | Who is most likely to get infected from SARS CoV-2? | (32) 10.32 | **(57) 18.39** | **(167) 53.87** | 00 | **00** | (54) 17.42 | |
| | How does SARS CoV-2 spread? | (84) 27.1 | (27) 8.71 | 00 | (30) 9.68 | **(152) 49.03** | (17) 5.48 | |
| | What would you do if you find someone coughing with high fever in your locality? | (71) 22.9 | **(216) 69.68** | (06) 1.94 | (5.48) 5.48 | | | |
| | What are the preventive measures WHO has imposed to minimise SARS Cov-2 spread? | (03) 0.97 | 00 | 00 | (01) 0.32 | **(305) 98.9** | 00 | (01) 0.32 |
| | How frequently should you wash or dispose of your face mask? | **(183) 59.03** | (05) 1.61 | **(68) 21.94** | (37) 11.94 | (17) 5.48 | | |
| | How often do you touch your face without washing or sanitising your hand? | **(70) 22.58** | (66) 21.3 | (29) 9.35 | (32) 10.32 | (113) 36.45 | | |
| | How safe is the clinic you work in terms of patients' exposure, having symptoms of coronavirus, and red eye? | **(29) 9.35** | (66) 21.3 | (58) 18.71 | **(109) 35.16** | **(48) 15.48** | | |
| | What should elderly people do in the period of lockdown? | (19) 6.13 | **(283) 91.29** | (01) 0.32 | (02) 0.65 | (05) 1.61 | | |
| | Eye Care Practitioners should use PPE during the COVID-19 pandemic. To what extent do you agree? | **(62) 20** | (111) 35.8 | (12) 3.88 | (125) 40.32 | | | |

Correct answers are given in bold. The number of possible responses differed between questions. The number of responses is given in brackets, with percentages below. The order of questions given here, separating questions assessing knowledge and awareness, differs to those presented to participants. Please see Supplemental Material for the full questions as presented to participants with answers.

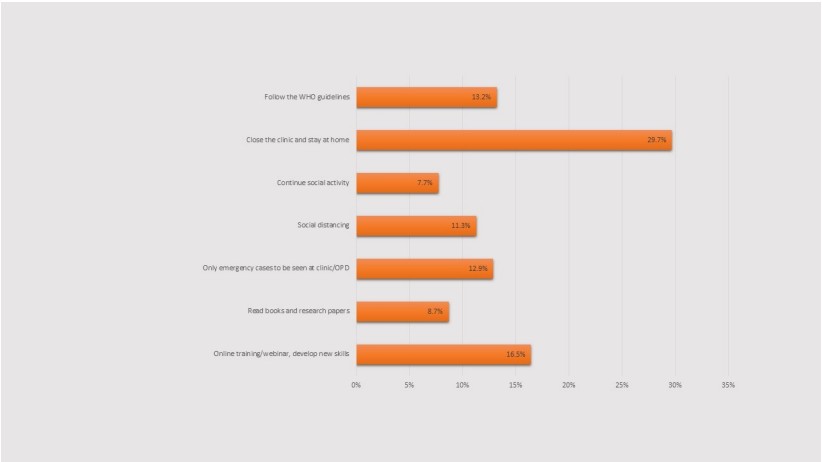

**Fig 1. What should ophthalmic professionals do during lockdown in Nepal?.**

cases within the country according to the WHO [20]. The Government of Nepal, in partnership with the WHO, is working tirelessly in preventing and controlling the disease. Building awareness around hygiene measures (hand wash procedure, respiratory hygiene, etc.) and social distancing have been given priority. Despite these evidence-based messages, there has been a considerable amount of misinformation and myths circulating within the country.

Within Nepal, the majority of eye health personnel fall within the 25–35 age bracket. Although the proportion of males and females within the eye health workforce nationally is unknown, a recent study from the Eastern Region of Nepal reported that approximately 55% of general health practitioners were female [21]. Within our study, the age range of our respondents was as expected, although with 55% of respondents being male, women may be slightly under-represented within this sample if the gender distribution for the eye health workforce is similar to that of general health practitioners; male participants may have been, for example, more willing to complete the online survey for cultural reasons.

Our survey shows that there is still considerable ignorance amongst those questioned regarding COVID-19. Only 56% felt it mandatory to use PPE if one is working in the ophthalmology/optometry clinic, despite reports of fatalities of ophthalmic clinicians due to COVID-19 [18]. The Centre for Disease Control and Prevention (CDC) and WHO have created guidelines for the safe running of emergency care in an ophthalmic setting. They have emphasised that PPE must be mandatory for ophthalmic clinicians to reduce the risk of contracting COVID-19 [22]. The PPE in the guidelines includes essentials to cover the mouth, nose (N95 mask preferably), eyes (goggles), face (visors), hands (gloves), and breath shields attached to the slit lamp. An example of PPE as worn by an ophthalmic assistant in the Eastern Region of Nepal, during the COVID-19 pandemic is illustrated in Fig 2. These are essential to restrict the spread of the virus from one person to another within the ophthalmic setup. Half of the participants were not aware that the detection of novel SARS Cov-2 was confirmed by RT-PCR [23]. One third of the participants felt that drinking hot water/ginger-garlic mix/turmeric/vitamin C soup, etc. kills the virus directly or indirectly. There is no evidence to support this statement, which has been frequently circulating around social media as misinformation [24, 25].

Approximately half of the participants were unaware of the lifespan of the virus on different surfaces [13, 14]. Knowledge of this can help clinicians to appropriately disinfect surfaces and equipment. Several participants were not sure if they should clean their masks daily, while others did not wear masks, and a few washed it once every two days. The WHO recommends

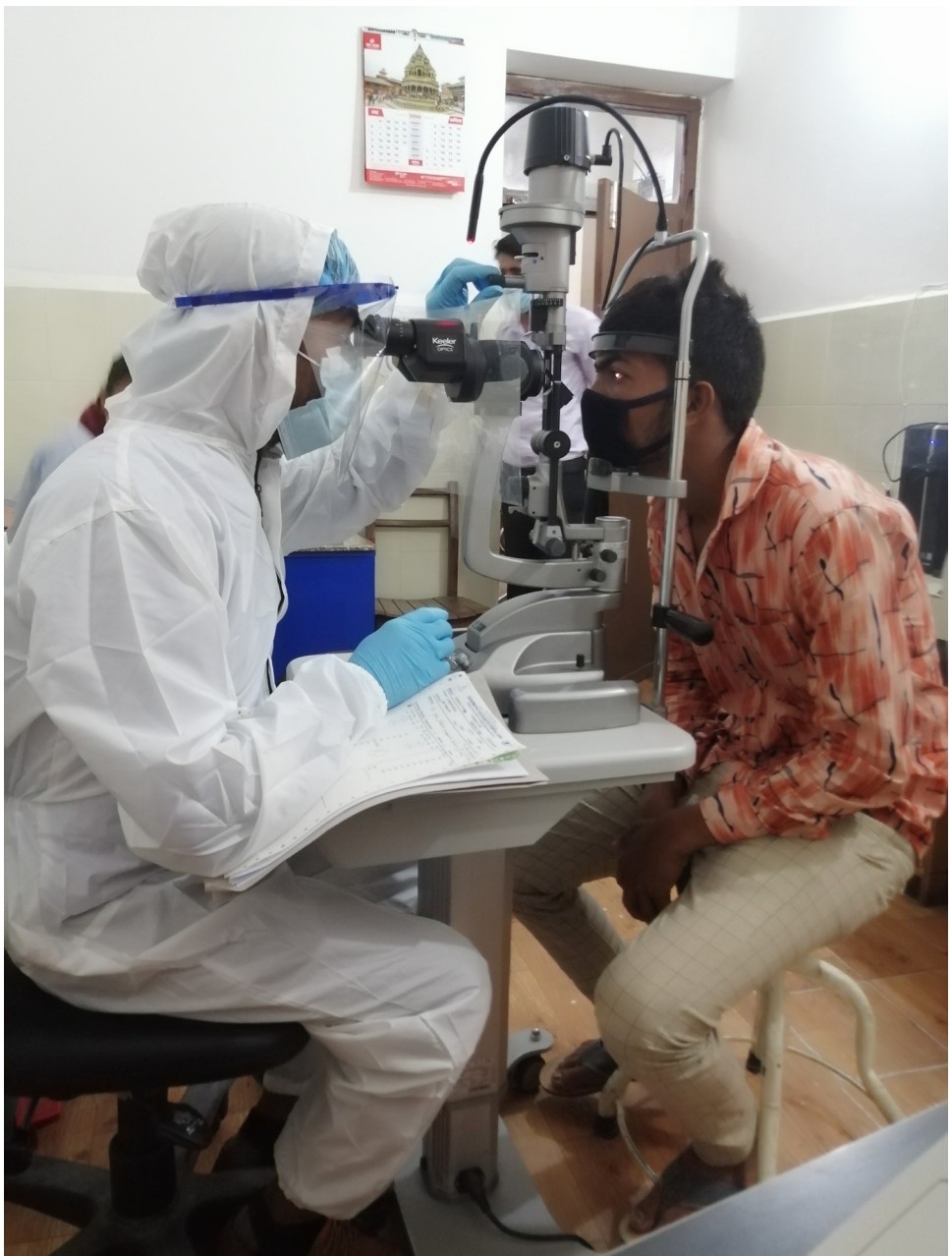

**Fig 2. An example of PPE as worn by an ophthalmic assistant in the Eastern Region of Nepal, during the COVID-19 pandemic.**

using a mask whilst in clinic, and importantly, there is a high risk of the infection being transmitted if the mask is not disposed of or sterilised properly on a daily basis, especially in susceptible areas [15, 26].

The CDC and WHO recommends at least 20 seconds should be spent while washing hands [27]. Most participants said they follow this recommendation.

The CDC also recommends 70% ethanol to be used as a multipurpose disinfectant in ophthalmology clinics and other surface cleaning activities [23]. More than half of participants

(59.0% (n = 183)) believed other less-potent disinfectants were adequate, which highlights an area that can be improved on through further education of clinic staff.

According to the American Academy of Ophthalmology clinical guidelines, cases of conjunctivitis have been reported [28]. As conjunctival secretions and tears from infected patients my contain virus RNA and those with conjunctival symptoms may pose higher risk [29]. In a recent study, of the total reported cases of ocular manifestations of COVID-19, only 0.8% had conjunctivitis; other coronaviruses have also been implicated in causing conjunctivitis, albeit rarely, with similar rates of conjunctivitis reported in previous studies [12, 30, 31]. Clinicians therefore need to be very cautious while examining these patients as there is a risk, they may be sources of infection of SARS-CoV-2. Knowledge of this is therefore important to ophthalmic practitioners. The majority of the participants knew that conjunctivitis was the most common ocular manifestation of COVID-19, which is reassuring and has not changed over some time. Different types of conjunctivitis and keratoconjuctivitis are the most frequently reported ocular manifestations, with one case-report of a patient demonstrating retinal changes detected by ocular coherence tomography [32–34].

As a knowledge assessment on ocular emergency within the context of COVID-19, 5% of the participants felt blurred vision as an emergency, which in our view unless acute in onset, would not be classed as such. There is therefore a chance that routine patients were being seen unnecessarily during lockdown by a minority of clinicians.

A free text opinion was taken from the participants on what should be done by a professional during lockdown, to get an idea on personal behaviour and frustrations they may feel. There are reports of people involved in domestic violence and suicide in the community due to limited social interaction. The responses to this question are given in Fig 1. Interestingly, the majority felt that they should not be working and should self-isolate at home. At the other end of the spectrum, 7.7% still felt that social activities should continue. This suggests their opinions are somewhat divided.

Approximately half of participants believed that only immunocompromised individuals and/or elderly people are susceptible to the SARS Cov-2, whereas half recognised that anybody can be infected, which has been stated by WHO [35]. There were mixed opinions regarding the spread of SARS-CoV-2, with approximately 10% of participants feeling that SARS-CoV-2 is transmitted as an aerosol. This mirrors the literature, as it still remains unclear exactly how the virus spreads within the environment. A study of aerosols did not find any contamination of air with SARS-CoV-2 in an isolation ward of COVID-19 patients [13].

Most of the participants surveyed had their clinics equipped with safety measures as advised by the WHO and American Academy of Ophthalmology, which recommends clinicians in ophthalmology clinics should use PPE [36]. This is reassuring, and indeed may be better than in settings in other countries where there has often been a challenge regarding adequate supplies of PPE.

There are some potential limitations to this study. Although we selected participants at random from the main eye hospitals in Nepal, only just over half of all those invited responded to the survey. This may cause a degree of selection bias. Furthermore, the COVID-19 pandemic is a rapidly evolving situation, which means that some of the "correct" answers to our survey that were accurate at the time of questioning, may have changed by the time of analysis. The strengths of this study are that this is the only such study to have been conducted in Nepal to the best of our knowledge, conducted within a short timeframe.

Further work must be carried out by employers, as well as local and national governments in Nepal, in order to prevent the spread of misinformation, and support practitioners to carry out eye care during this pandemic safely. The views expressed in this study reflect those of educated professionals: it is therefore likely that public awareness is less, with belief in rumours is

greater, than what we have identified. Additional studies are necessary to accurately assess the opinions of the general public on COVID-19 in Nepal.

## Conclusion

Our study shows that knowledge amongst eye care practitioners is classed as only satisfactory. We therefore recommend ophthalmic practitioners to enhance and develop this knowledge as a priority. There are abundant, free webinars on COVID-19 and eye care from which individuals can benefit. The views expressed within the awareness questions are concerning, as some of these responses highlight a degree of misinformation, which may be of concern for institutions and local government.

With rumours spreading quickly during crises, there has been an increasing amount of "fake news" or misinformation circulating in Nepal during this pandemic, and this study highlights some of these issues. It is crucial that clinicians remember to check the scientific evidence before altering their practice or recommending interventions to their patients.

## Supporting information

**S1 Data.**
(DOCX)

## Acknowledgments

Authors thank staff at Sagarmatha Choudhary Eye Hospital for their support in every aspect of the study. Special thanks to Mr Abhishek Roshan and Dr Sanjay Kumar Singh for facilitating this study during the lockdown period.

## Author Contributions

**Conceptualization:** Sandip Das Sanyam, Pankaj Chaudhary.

**Data curation:** Sandip Das Sanyam, Sanjay Kumar Sah, Pankaj Chaudhary.

**Formal analysis:** Sandip Das Sanyam, Sanjay Kumar Sah, Jeremy J. Hoffman.

**Funding acquisition:** Matthew J. Burton.

**Investigation:** Sandip Das Sanyam, Sanjay Kumar Sah.

**Methodology:** Sandip Das Sanyam, Sanjay Kumar Sah, Pankaj Chaudhary, Matthew J. Burton.

**Project administration:** Sandip Das Sanyam, Pankaj Chaudhary.

**Resources:** Matthew J. Burton.

**Software:** Sandip Das Sanyam.

**Supervision:** Sandip Das Sanyam, Sanjay Kumar Sah, Matthew J. Burton, Jeremy J. Hoffman.

**Validation:** Sandip Das Sanyam, Sanjay Kumar Sah.

**Visualization:** Sandip Das Sanyam, Pankaj Chaudhary, Matthew J. Burton.

**Writing – original draft:** Sandip Das Sanyam, Sanjay Kumar Sah, Pankaj Chaudhary, Jeremy J. Hoffman.

**Writing – review & editing:** Sandip Das Sanyam, Sanjay Kumar Sah, Pankaj Chaudhary, Matthew J. Burton, Jeremy J. Hoffman.

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
