## [Decision Letter · Decision Letter 0]

5 Mar 2021

PONE-D-20-29438

Knowledge and awareness-based survey of COVID-19 1withinthe eye care professionin Nepal:2misinformation is hiding the truth.

PLOS ONE

Dear Dr. Sanyam,

Thank you for submitting your manuscript to PLOS ONE. After careful consideration, we feel that it has merit but does not fully meet PLOS ONE’s publication criteria as it currently stands. Therefore, we invite you to submit a revised version of the manuscript that addresses the points raised during the review process.

The manuscript has been evaluated by three reviewers, and their comments are available below.

The reviewers have raised a number of concerns that need attention. They request additional information on methodological aspects of the study (such as the inclusion of information on the sample size and patient population), revisions to the statistical analyses, and they request additional attention to the interpretation of the study's results in the context of the current literature.

Could you please revise the manuscript to carefully address the concerns raised?

We look forward to receiving your revised manuscript.

Kind regards,

Marianne Clemence

Associate Editor

PLOS ONE

Journal Requirements:

2)  Please provide additional details regarding participant consent. In the ethics statement in the Methods and online submission information, please ensure that you have specified (1) whether consent was informed and (2) what type you obtained (for instance, written or verbal, and if verbal, how it was documented and witnessed). If your study included minors, state whether you obtained consent from parents or guardians. If the need for consent was waived by the ethics committee, please include this information.

3) Thank you for including your ethics statement:  "Ethical permission was obtained".   

4) Thank you for stating the following in your Competing Interests section: 

[nil].

5) We note that you have indicated that data from this study are available upon request. PLOS only allows data to be available upon request if there are legal or ethical restrictions on sharing data publicly. For information on unacceptable data access restrictions, please see http://journals.plos.org/plosone/s/data-availability#loc-unacceptable-data-access-restrictions.

6) Please amend either the title on the online submission form (via Edit Submission) or the title in the manuscript so that they are identical.

7)  We noticed you have some minor occurrence of overlapping text with the following previous publication(s), which needs to be addressed:

https://onlinelibrary.wiley.com/doi/abs/10.1002/jmv.25725

https://www.cehjournal.org/article/understanding-covid-19-and-the-virus-that-causes-it/

In your revision ensure you cite all your sources (including your own works), and quote or rephrase any duplicated text outside the methods section. Further consideration is dependent on these concerns being addressed.

Reviewers' comments:

Reviewer's Responses to Questions

**Comments to the Author**

1. Is the manuscript technically sound, and do the data support the conclusions?

Reviewer #1: Partly

Reviewer #2: Partly

Reviewer #3: Yes

2. Has the statistical analysis been performed appropriately and rigorously? 

Reviewer #1: No

Reviewer #2: I Don't Know

Reviewer #3: Yes

3. Have the authors made all data underlying the findings in their manuscript fully available?

Reviewer #1: Yes

Reviewer #2: Yes

Reviewer #3: Yes

4. Is the manuscript presented in an intelligible fashion and written in standard English?

Reviewer #1: Yes

Reviewer #2: Yes

Reviewer #3: Yes

5. Review Comments to the Author

Reviewer #1: Review report

Manuscript Number: PONE-D-20-29438

Manuscript Title: Knowledge and awareness-based survey of COVID-19 1within the eye care profession in Nepal:2 misinformation is hiding the truth.

1. The paper is about a contemporary issue in public health, it is clearly written, and the topic is of high importance. The abstract is clear. Title is ambiguous.

2. The introduction should give background to the choice of study participants for the study, should demonstrate gap in knowledge, justification for the study and show relevance of the research to policy and practice. Why eye care practitioners? Why is the study relevant? Will it help to better control the emergency? All these did not appear clearly in the introduction.

3. The method: Authors should state who commissioned the study. What IRB granted approval, approval date and number (see line 88). It is important to state how the sample size for the study was determined, how instrument for data collection was checked for validity and reliability.

4. The choice of analytical approach is too basic. There is no explanation of the independent and dependent variables. There is need to conduct inferential statistics, compute mean scores, standard deviations and ratios to better explain the knowledge and awareness scores obtained.

5. It is always better to avoid starting a sentence with a figure. See lines 124,128,136.

6. The References are recent and relevant, but some are incomplete. Authors may want to update their references with related publications on the subject matter.

7. Recommendation for further study appeared in the conclusion, it may be better situated in the discussion section.

Reviewer #2: 1. Financial disclosure and competing interests should be edited according the journal’s instructions.

2. IRB’s full name and committee number should be mentioned.

3. Authors should include whether the inform consent was given orally or in a written form.

4. Authors should consider emphasizing the specific importance of PPE and supplementary measures (e.g. clear plastic boards placed on the slit lamp) utilization in the eye clinic.

5. Authors should update their data and cite contemporary studies regarding ocular transmission of SARA-CoV-2 and ocular manifestations. They should try and compare it to what was known in the time the questionnaires were filled and try to explain, in their opinion, if the participants would answer differently in light of the current data.

6. If available, it would be helpful to know additional background data of the participants (e.g. size of clinic, number of patients seen in a certain amount of time, etc.).

7. Authors should mention and regard the fact that the majority of participants were quite of a homogenous group and of relatively young age. They should elaborate on it in their discussion.

8. In continuum to the previous comment, authors should try and explain their results in light of the cultural and occupational background of the participants.

Reviewer #3: accepted with no comments. the manuscript is well written. The manuscript describe a technically sound piece of scientific research with data that supports the conclusions. the manuscript discussed an important issue and it seems quite interesting that most of the participants are not aware enough about this pandemic.

6. PLOS authors have the option to publish the peer review history of their article (what does this mean?). If published, this will include your full peer review and any attached files.

Reviewer #1: No

Reviewer #2: No

Reviewer #3: No

---

## [Author Response · Author response to Decision Letter 0]

31 Mar 2021

Reviewer #1: 

1. The paper is about a contemporary issue in public health, it is clearly written, and the topic is of high importance. The abstract is clear. Title is ambiguous.

Thank you very much for reviewing our manuscript and for providing valuable feedback. Thank you for appreciating the importance of this topic. Regarding the title, we have amended the title on the online system so that there is no confusion.

2. The introduction should give background to the choice of study participants for the study, should demonstrate gap in knowledge, justification for the study and show relevance of the research to policy and practice. Why eye care practitioners? Why is the study relevant? Will it help to better control the emergency? All these did not appear clearly in the introduction.

Thank you for this comment. On a pilot online questionnaire of the same questions among 10 participants before beginning the study, the knowledge and awareness seemed poor. We therefore felt there was a need to conduct a larger, more thorough study to check if the perception was the same or different across the entire nation. Eye care practitioners are at close contact with patients and there is a high risk of transmitting COVID-19 virus through aerosols between patients and eye care personnel.

Dr Li, an ophthalmologist from Wuhan who unfortunately died from COVID-19 early during the pandemic, had said COVID-19 needs to be taken seriously by eye care workers in order to protect themselves. References to this have now been added to the introduction.

If gaps in knowledge were found, then these could be addressed through targeted education / training which helps reduce the risk to healthcare workers and their patients and promote good infection control practices. 

The introduction has now been updated to reflect these points and now reads as follows:

During this period, there have been many rumors and misinformation on COVID-19 emerging, which might mislead ophthalmic personnel. 

Dr Li Wenliang from Wuhan, who died from COVID-19, was an ophthalmologist who warned about the contagious nature of this virus and had asked eye care practitioners worldwide to take safety precautions, given the close proximity eye care practitioners are in with their patients. The aerosols generated from diseased patients attending for eye examinations have a higher tendency to infect eye care personnel if the appropriate protection is lacking. A pilot study of ours using the same study tool revealed poor knowledge and awareness within the eye care practitioners at our institution. The aim of this study was to review the knowledge and awareness of COVID-19 amongst eye care personnel in Nepal. If gaps in knowledge were found, then this could be addressed through targeted education and training, which will help reduce the risk to healthcare workers and promote good infection control practices.

3. The method: Authors should state who commissioned the study. What IRB granted approval, approval date and number (see line 88). It is important to state how the sample size for the study was determined, how instrument for data collection was checked for validity and reliability.

Thank you for these comments. As noted in the response to the Editor, we have updated the ethics statement within the methods section to include the details of the ERB, approval date and number. Please see the response above which includes the updated ethics statement.

Regarding the sample size, we have added the following text to the manuscript to address the logic behind how this was determined. This section now reads as follows:

The survey was sent to 600 of these clinicians working in Nepal. Participation was anonymous and voluntary. Representatives from a random selection of 12 of the 18 secondary or tertiary eye hospitals and clinics across the spectrum of health facilities within Nepal were contacted and asked to select 50 individuals randomly at their institution and contacts who were involved in eye care practice during COVID-19, who were then invited to take the survey after their agreement. This random sample of 600 ophthalmic clinicians (approximately 40% of eye care professionals within Nepal) was calculated to be representative of the staff involved in eye care, whilst being practical enough to implement, making use of established networks between eye care providers. 

Regarding the validity, online forms could only be completed by those directly invited and values were entered directly by the individual. This has been clarified within the text and this section now reads as follows:

In order to ensure validity, online forms could only be completed by those directly invited and values were entered directly by the individual.

4. The choice of analytical approach is too basic. There is no explanation of the independent and dependent variables. There is need to conduct inferential statistics, compute mean scores, standard deviations and ratios to better explain the knowledge and awareness scores obtained.

Thank you for this observation. This study was performed as a survey of the views of ophthalmic professionals, and the results reported in the knowledge section represent the percentage of question that our sample correctly answered. Therefore, reporting the results in this way, it is not possible to do any further statistical analysis. We have, however, clarified the mean results of all participants, and given the standard deviation for these results. We have only reported on the mean overall results for the Knowledge component of the questionnaire. We have not reported the mean correct results for the Awareness section, as for these questions there is no correct response.

Regarding the comment about independent and dependent variables, in this survey we are not establishing cause and effect; as such we do not have any dependent or independent variables to define.

We have updated the results section of the text with the following, to include standard deviations as suggested:

The calculated mean knowledge performance score was 69.65% (SD ± 22.81), falling into the “satisfactory” category.

5. It is always better to avoid starting a sentence with a figure. See lines 124,128,136.

This has been corrected, thank you for alerting us to this.

6. The References are recent and relevant, but some are incomplete. Authors may want to update their references with related publications on the subject matter.

This has been updated now, thank you.

7. Recommendation for further study appeared in the conclusion, it may be better situated in the discussion section.

Shifted the recommendation section within the discussion, thank you.

Reviewer #2: 1. Financial disclosure and competing interests should be edited according to the journal’s instructions.

We have edited as per PLOS ONE requirements, thank you.

2. IRB’s full name and committee number should be mentioned.

We have now mentioned this within the method section, thank you.

3. Authors should include whether the inform consent was given orally or in a written form.

Clarified the same in the method section of the manuscript, thank you.

4. Authors should consider emphasizing the specific importance of PPE and supplementary measures (e.g. clear plastic boards placed on the slit lamp) utilization in the eye clinic.

We have revealed the importance of breath shields and other PPE for eye care practitioner in the discussion section, thank you.

5. Authors should update their data and cite contemporary studies regarding ocular transmission of SARA-CoV-2 and ocular manifestations. They should try and compare it to what was known in the time the questionnaires were filled and try to explain, in their opinion, if the participants would answer differently in light of the current data.

Thank you for this comment. We have updated this in the discussion section.

6. If available, it would be helpful to know additional background data of the participants (e.g. size of clinic, number of patients seen in a certain amount of time, etc.).

Thank you for this suggestion. We have added this to the results section. The following sentence has been added:

These eye health personnel were from a diverse range of health facilities, from small single-practitioner clinics to stand-alone tertiary referral eye hospitals.

7. Authors should mention and regard the fact that the majority of participants were quite of a homogenous group and of relatively young age. They should elaborate on it in their discussion.

Thank you for raising this point. This is now explained within the discussion and currently reads as:

Within Nepal, the majority of eye health personnel fall within the 25-35 age bracket. Although the proportion of males and females within the eye health workforce nationally is unknown, a recent study from the Eastern Region of Nepal reported that approximately 55% of general health practitioners were female. [20] Within our study, the age range of our respondents was as expected, although with 55% of respondents being male, women may be slightly under-represented within this sample if the gender distribution for the eye health workforce is similar to that of general health practitioners; male participants may have been, for example, more willing to complete the online survey for cultural reasons.

8. In continuum to the previous comment, authors should try and explain their results in light of the cultural and occupational background of the participants.

Thank you for this comment. We have addressed this within our response to point No. 7 above.

Reviewer #3: accepted with no comments. the manuscript is well written. The manuscript describe a technically sound piece of scientific research with data that supports the conclusions. the manuscript discussed an important issue and it seems quite interesting that most of the participants are not aware enough about this pandemic.

Thank you for your review.

6. PLOS authors have the option to publish the peer review history of their article (what does this mean?). If published, this will include your full peer review and any attached files.

Do you want your identity to be public for this peer review? For information about this choice, including consent withdrawal, please see our Privacy Policy.

Reviewer #1: No

Reviewer #2: No

Reviewer #3: No

---

## [Editor Report · Decision Letter 1]

9 May 2021

PONE-D-20-29438R1

Knowledge and awareness-based survey of COVID-19 within the eye care profession in Nepal: misinformation is hiding the truth

PLOS ONE

Dear Dr. Sanyam,

Thank you for submitting your manuscript to PLOS ONE. After careful consideration, we feel that it has merit but does not fully meet PLOS ONE’s publication criteria as it currently stands. Therefore, we invite you to submit a revised version of the manuscript that addresses the points raised during the review process.

We look forward to receiving your revised manuscript.

Kind regards,

Bernadine Nsa Ekpenyong, PhD

Academic Editor

PLOS ONE

Journal Requirements:

Additional Editor Comments (if provided):

The authors have not addressed the analysis section of the review comments. both reviewer 1 and 2 expressed the need for the authors to do further analysis to express demographic, social and occupational differences in knowledge score. the authors response on this is not acceptable.

it is also not clear whether the instrument for data collection was adapted, or how it was developed. if the later is the case, how issues around reliability and validity of instrument was addressed.

---

## [Author Response · Author response to Decision Letter 1]

21 Jun 2021

Journal Comments: 

Please review your reference list to ensure that it is complete and correct. If you have cited papers that have been retracted, please include the rationale for doing so in the manuscript text or remove these references and replace them with relevant current references. Any changes to the reference list should be mentioned in the rebuttal letter that accompanies your revised manuscript. If you need to cite a retracted article, indicate the article’s retracted status in the References list and also include a citation and full reference for the retraction notice.

Reference list is now updated, and URLs are checked, thank you.

Additional Editor Comments (if provided):

The authors have not addressed the analysis section of the review comments. both reviewer 1 and 2 expressed the need for the authors to do further analysis to express demographic, social and occupational differences in knowledge score. the authors response on this is not acceptable.

Thank you for this comment. We have now run further analysis on the knowledge results in terms of the background demographics (gender, age group and occupation). Fisher’s exact analysis does not show any significant difference for these demographic variables and the knowledge score obtained. We have therefore added the following sentence to the results section:

We found no evidence to suggest an association between demographic variables (specifically age, gender and occupation) and answering the knowledge questions correctly (data not shown).

We have also added the following to the methods section to describe the statistical testing performed:

Fisher’s exact testing was performed to look for significant associations between categorical variables.

it is also not clear whether the instrument for data collection was adapted, or how it was developed. If the latter is the case, how issues around reliability and validity of instrument was addressed.

Thank you for highlighting this. We have added the following now to the methods section to describe how the questionnaire was developed:

We used qualitative methods to generate the questions on knowledge and awareness. This has been shown to enrich the quality of questionnaire items and to improve the content validity of the questionnaire. Questions developed in this way derives item generation from the population of interest, rather than that of the researchers. Focus group discussion took place between eye care professionals, statisticians, and microbiologists, facilitated by the investigators.

Note:

Please be informed that we have also added one more author cum supervisor to the manuscript.

---

## [Editor Report · Decision Letter 2]

5 Jul 2021

Knowledge and awareness-based survey of COVID-19 within the eye care profession in Nepal: misinformation is hiding the truth

PONE-D-20-29438R2

Dear Dr. Sanyam Sandip Das

We’re pleased to inform you that your manuscript has been judged scientifically suitable for publication and will be formally accepted for publication once it meets all outstanding technical requirements.

Kind regards,

Bernadine Nsa Ekpenyong, PhD

Guest Editor

PLOS ONE
---

## [Editor Report · Acceptance letter]

13 Jul 2021

PONE-D-20-29438R2 

Knowledge and awareness-based survey of COVID-19 within the eye care profession in Nepal: misinformation is hiding the truth 

Dear Dr. Sanyam:

I'm pleased to inform you that your manuscript has been deemed suitable for publication in PLOS ONE. Congratulations! Your manuscript is now with our production department. 

Kind regards, 

on behalf of

Dr. Bernadine Nsa Ekpenyong 

Guest Editor

PLOS ONE